# Decoding Safety Feedback from Diverse Raters: A Data-driven Lens on Responsiveness to Severity

**Pushkar Mishra**[1], **Charvi Rastogi**[1], **Stephen R. Pfohl**[2], **Alicia Parrish**[1], **Tian Huey Teh**[1], **Roma Patel**[1], **Mark Diaz**[2], **Ding Wang**[2], **Michela Paganini**[1], **Vinodkumar Prabhakaran**[2], **Lora Aroyo**[1], **Verena Rieser**[1]

[1] *Google DeepMind*
[2] *Google Research*

Corresponding authors: `{pushkarmishra,charvir,spfohl}@google.com`

Reviewed on OpenReview: `https://openreview.net/forum?id=vx6qrM7VB5`

## Abstract

Ensuring the safety of Generative AI requires a nuanced understanding of pluralistic viewpoints. In this paper, we introduce a novel data-driven approach for analyzing ordinal safety ratings in pluralistic settings. Specifically, we address the challenge of interpreting nuanced differences in safety feedback from a diverse population expressed via ordinal scales (e.g., a Likert scale). We define non-parametric *responsiveness metrics* that quantify how raters convey broader distinctions and granular variations in the severity of safety violations. Leveraging publicly available datasets of pluralistic safety feedback as our case studies, we investigate how raters from different demographic groups use an ordinal scale to express their perceptions of the severity of violations. We apply our metrics across violation types, demonstrating their utility in extracting nuanced insights that are crucial for aligning AI systems reliably in multi-cultural contexts. We show that our approach can inform rater selection and feedback interpretation by capturing nuanced viewpoints across different demographic groups, hence improving the quality of pluralistic data collection and in turn contributing to more robust AI alignment.

## 1 Introduction

Ensuring the safety of Generative AI is paramount for their responsible deployment and societal trust. Safety evaluation tasks that use *binary ratings*, such as *safe* and *unsafe*, lack the granularity needed for effective alignment with human preferences (Wu et al., 2023; Collins et al., 2024). To capture more fine-grained perceptions of the severity of potential harm, safety alignment tasks increasingly employ *ordinal scales*, such as Likert scales (Curry et al., 2021), which allow for more nuanced feedback that is vital to the creation of alignment datasets for Supervised Fine-Tuning (SFT) / Reinforcement Learning from Human Feedback (RLHF). A significant challenge, however, is that these scales are harder to interpret due to variations in individual perceptions and response biases (Paulhus, 1991) such as extreme responses (Greenleaf, 1992), central tendency (for odd scales), and forced choice and polarization (for even scales). Such biases can propagate to datasets and optimization objectives for safety alignment (Kaufmann et al., 2024), leading to exaggerated safety behaviors in Generative AI, e.g., false refusal of benign requests.

Yet another significant challenge that emerges from recent research is that safety perceptions are not uniform; they vary significantly across individuals and groups (Aroyo et al., 2024; Kirk et al., 2024; Rastogi et al., 2024). Poor understanding of differences in how diverse raters or rater groups perceive the severity of violations and map the perceived severity to ordinal scores can lead to the design of optimization objectives that are insensitive or even harmful to certain cultures. Moreover, when LLMs trained on such *pluralistic*

ordinal ratings are used as judges or reward models (Bavaresco et al., 2024; Wang et al., 2024), they may not represent well the granular variations or even broader distinctions in safety perceptions of different raters or rater groups. Both cases cause downstream harms due to poor understanding of nuanced safety feedback.

This paper introduces a novel data-driven *non-parametric* approach for analyzing nuanced pluralistic safety feedback, offering deeper and more robust insights than traditional non-parametric and parametric approaches. Leveraging two publicly available datasets of pluralistic safety feedback as our case studies, we address the challenge of understanding the differences in ordinal ratings from diverse demographic groups when expressing the severity of safety violations. The main contributions of this paper are:

**Metrics development:** We define robust *responsiveness metrics* from observable data to analyze the feedback of individual raters or rater groups for varying levels of severity. These metrics allow us to:

1. **Measure responsiveness to severity**: How do different raters or rater groups use a given ordinal scale to score the varying levels of severity of violations?

2. **Compare responsiveness**: Do different raters or rater groups respond similarly to varying levels of severity of violations?

**Metrics application:** We apply these metrics to two publicly available pluralistic datasets of safety feedback, demonstrating their utility in extracting nuanced insights that are crucial for aligning AI systems in multi-cultural contexts, specifically:

- **Understanding scale usage**: uncovering patterns in the use of the given Likert scale, thus elucidating pluralistic variations in the expressions of demographic groups;

- **Capturing nuanced viewpoints**: quantifying the responsiveness of demographic groups to severity across violation types, resulting in a deeper understanding of pluralistic viewpoints;

- **Informing pluralistic data collection**: establishing a reliable and repeatable process for selecting raters with high responsiveness to severity from different groups;

- **Guiding safety alignment in pluralistic settings**: reflecting the granular variations in safety perceptions of diverse raters alongside the broader distinctions in their safety perceptions.

## 2 Related work

Our work builds on the state-of-the-art in eliciting nuanced human feedback in Generative AI evaluation and expands existing research on understanding of human judgments.

**Nuanced human feedback for AI safety.** Collecting human perspectives on behavior of generative AI models is exceedingly commonplace with growth in its usage in real world tasks. Across the literature on AI evaluation, different configurations of human feedback have been studied, with an outsized focus on binary (0/1) human feedback. Recent research (Collins et al., 2024; Arhin et al., 2021; Denton et al., 2021) discusses the limitations of binary feedback in capturing the nuance involved in generative AI evaluation, especially in safety. Wu et al. (2023); Collins et al. (2024) propose fine-grained human feedback encompassing evaluation across multiple attributes and with higher density, yielding improvement in downstream AI tasks via RLHF. Further, Rauh et al. (2024); Jiang et al. (2021) emphasize the importance of measuring extent of harm (severity) in evaluation of algorithms. Another dimension in collecting human feedback relates to the identity of the human providing the feedback. The role of rater identity in their annotation has been discussed extensively in AI evaluation literature (Denton et al., 2021; Arhin et al., 2021; Aroyo et al., 2024; Homan et al., 2023; Pei & Jurgens, 2023; Davani et al., 2024). For developing AI that aligns with human values, Sorensen et al. (2024) show the importance of considering pluralistic viewpoints from a diverse set of raters. Our research builds upon this body of work by specifically examining human feedback collected on a fine-grained ordinal scale from raters belonging to different groups with different collective identities.

**Interpretation and calibration of human judgments.** Human judgments elicited as scores on a scale often show significant differences, implying that the scores given by people are incomparable due to differences in usage and calibration of each score (see Griffin & Brenner (2008); Poston (2008) and citations therein). Such differences in human scores are usually interpreted through simplifying modeling assumptions about

how miscalibrations present in the data. These modeling assumptions include linear models with additive biases corresponding to rater identity (Bürkner & Vuorre, 2019; Paul, 2011; Barr et al., 2013), models with rater identity-based scale-and-shift biases (Paul, 2011; Roos et al., 2011), mixed-effects models, among others (Wang & Shah, 2019). However, research has shown that issues of human judgment calibration are often more complex, causing significant violations to these simplified assumptions (see Griffin & Brenner (2008) and citations therein). In this work, while making minimal assumptions on the nature of miscalibration in human judgments, we provide *non-parametric* metrics to quantify the consistency of raters in reflecting varying levels of severity. Traditional non-parametric metrics like Kendall's $\tau$ and area under the PR or ROC curves do not capture well the responsiveness to severity. Using real-world datasets, we surface insights from our proposed metrics into the feedback patterns of different rater groups.

## 3 Setup

We consider a general setup with two different rater populations that reflect two contrasting safety feedback paradigms (Rottger et al., 2022): crowd raters who *indicate* safety preferences of a diverse population, and trained raters who follow detailed guidelines to *prescribe* what is safe or unsafe.

1. *Crowd raters* provide pluralistic safety feedback using an ordinal scale, with each rating representing the perception of a certain rater group. Each crowd rater reviews the given item and provides an integer score on a 0-$K$ Likert scale, where 0 is not harmful and $K$ is completely harmful.

2. *Trained raters* strictly follow a set of prescribed guidelines. For each item, they provide a binary score of 0 or 1 indicating their feedback of safe or unsafe respectively.

### 3.1 Data model

To formalize our assumptions regarding the scores given by the individual raters, i.e., individual trained raters and individual crowd raters, we define a data model. Let $V_i$ be the *true underlying severity* of item $i$. Let $V_i^j$ be the true severity of item $i$ as perceived by rater $j$. Then,

$$V_i^j = f_j(V_i) \tag{1}$$

where $f_j$ is the rater-specific perception function. This model acknowledges that true underlying severity $V$ of items is not directly measurable but is still the principal factor that monotonically influences every rater's judgment. In the case of trained raters, $f_j$ represents the shared understanding of severity prescribed by their strict guidelines that serve as a framework to operationalize the theoretical notion of severity. In the case of crowd raters, $f_j$ represents some perception of true severity based on their lived experiences. In other words, perception of rater $j$ is primarily distributed along the dimension $V^j$ such that $V_i^j$ represents the unobservable true severity of item $i$ as perceived by rater $j$. Hereon, we use the notation $V^j$ for true severity as perceived by rater $j$ and the notation $V$ for the true underlying severity, or simply, true severity.

The relationship between $V_i^j$ and the scores $S_{ij}$ of the two rater populations can then be stated via response functions as:

$$M_{jk}(V_i^j) := P(S_{ij} \geq k | V_i^j) \tag{2}$$

where $M_{jk}(V_i^j)$ is non-decreasing in $V_i^j$. Further, we have that,

- for trained raters, $S_{ij} \in \{0, 1\}$, where $S_{ij} = 1$ with probability $M_{j1}(V_i^j)$.

- for crowd raters, $S_{ij} \in \{0, 1, \ldots, K\}$, where $S_{ij} = k$ with probability $M_{jk}(V_i^j) - M_{j(k+1)}(V_i^j)$ and $M_{j(K+1)}(V_i^j) = 0$.

We assume $V$ to be unidimensional following the standard assumption of a unidimensional latent trait found in frameworks like Item Response Theory (Samejima, 1968). We note that if $V$ is taken to be multidimensional, then our metrics can be applied per dimension provided that scores from the raters are also collected per dimension. But more importantly, unlike Item Response Theory (IRT), we do not assume some common latent calibrated scale(s) across all the raters, i.e., $V^j$ are specific to the raters, and we impose no further restrictions on them. This is crucial for pluralistic settings.

### 3.2 Responsiveness to severity

Having a Likert scale allows crowd raters to express their safety feedback on a severity spectrum rather than classifying items as safe or unsafe. However, Likert scales themselves do not guarantee that rater scores will meaningfully reflect the severity of violations. For example, there may be raters who only use the ends of the scale, or those who cluster all their feedback around certain scores. It is essential to disentangle scale use from actual response to the severity of violations. While one might ideally want to define responsiveness as a direct relationship between scores $S$ of a rater $j$ that capture $V^j$ and true severity $V$, this is not practically feasible. True severity $V$ is a latent construct that cannot be directly or exactly determined. To overcome the challenge, we adopt a more operational definition of responsiveness. We formally define responsiveness to the severity of violations as being composed of the following two properties that can be quantified using observable data:

- **Ability to stochastically order severity**. If a rater is responsive to the severity of violations, then a higher score from them should correspond to a higher probability of true severity $V$ crossing any threshold $T = t$. This is the principle of first-order stochastic dominance, which can be stated as $P(V > T|S = s_1, T = t) \geq P(V > T|S = s_2, T = t)$ for all $T \in \mathcal{T}$ when $s_1 > s_2$.

- **Ability to discriminate between distinct levels of severity**. If a rater is responsive to the severity of violations, then they should be able to discriminate the items whose true severities $V$ are above a given threshold $T = t$ from those below that threshold. This means that $P(S \geq s|V > T, T = t) \geq P(S \geq s|V \leq T, T = t)$ for all $T \in \mathcal{T}$.

Intuitively, these two properties together signify that a rater who is responsive to the severity of violations is able to convey both granular and broader variations in the true underlying severity $V$ using the Likert scale. Our approach to characterizing responsiveness to severity relies on a notion of stochastic ordering that is related to the classic decision-theoretic concepts of stochastic ordering and outcome monotonicity (Birnbaum & Navarrete, 1998). The stochastic ordering property can further be related to monotonicity of a calibration curve or reliability diagram (DeGroot & Fienberg, 1983) in the case that we consider calibration of crowd rater scores against some binarized reference for severity. The property concerning the ability of raters to discriminate between distinct levels of severity is related to the notion of discriminability from signal detection theory (McNicol, 2005) used to motivate the design of metrics such as the area under the receiver operating characteristic (ROC) curve and Kendall's $\tau$ (Kendall, 1938). Furthermore, the property is also related to the notion of discrimination between different levels of the latent trait in Item Response Theory (Samejima, 1968) and Mokken Scale Analysis (Mokken, 1971).

## 4 Metric design

Next, our aim is to quantify the responsiveness of crowd raters using Likert scales to the severity of violations in order to evaluate and compare them. Such a quantification can be achieved by individually quantifying the two properties that constitute responsiveness.

### 4.1 Reference for true severity

We note that during safety alignment, the notion of true severity $V$ is operationalized depending on the choice of the feedback paradigm. When safety alignment is conducted based on the feedback of trained raters, true severity $V$ is operationalized as captured by their guidelines, i.e., $V^g$. When safety alignment is conducted based on the feedback of crowd raters, true severity $V$ is operationalized as captured by the collective judgment of the crowd, i.e., $V^c$. In the latter case, there is no dependence on the existence of any guidelines. There can be other possible operationalizations too, but we focus on these two.

In any operationalization, true severity $V$ remains directly unobservable. We take $U$ to denote the observable binary reference such that $U = 1$ signals $V > T$ and $U = 0$ signals $V \leq T$. We can now reformulate the inequalities presented above for the ability to stochastically order and the ability to discriminate by leveraging $U = 1$ in place of $V > T$ and $U = 0$ in place of $V \leq T$. Note that since we do not assume a fixed threshold $T$, it is not straightforward to directly substitute $U = 1$ into the threshold-specific inequalities. However, we

show that if the two inequalities hold for all individual thresholds $T$, then they also hold for $U = 1$. Proofs for both the inequalities are in appendix A.

- **Ability to stochastically order**. We had that $P(V > T|S = s_1, T = t) \geq P(V > T|S = s_2, T = t)$ when $s_1 > s_2$, for all $T = t$. Hence, we can prove that $P(U = 1|S = s_1) \geq P(U = 1|S = s_2)$ when $s_1 > s_2$.

- **Ability to discriminate**. We had that $P(S \geq s|V > T, T = t) \geq P(S \geq s|V \leq T, T = t)$ for all $T = t$. Hence, we can prove that $P(S \geq s|U = 1) \geq P(S \geq s|U = 0)$.

Depending on the two operationalizations of true severity $V$ under consideration, we can obtain the observable binary reference $U$, i.e., *guideline-based* reference or *crowd-based* reference, as described:

1. We can treat the binary scores of trained raters as the binary reference $U$. We assign the individual binary scores of trained raters directly to the respective items. Here, $S$ and $T$ are trivially independent. When $U$ is obtained from trained raters, we are quantifying the responsiveness to severity $V^g$.
2. We can derive the binary reference $U$ from crowd raters themselves, excluding the rater(s) being evaluated to maintain non-circularity. Using points 1 to $K$ on the scale as exogenous boundaries to maintain the independence of $S$ and $T$, we obtain the binary reference $U$ per boundary by binarizing the individual Likert scores of crowd raters, excluding those being evaluated. When $U$ is obtained from crowd raters themselves, we are quantifying the responsiveness to severity $V^c$.

The design of our metrics itself is agnostic to the choice of the reference, and we work with both. References can have their own properties, so responsiveness must be interpreted in light of the chosen operationalization.

### 4.2 Metrics for the two properties

We formulate metrics for the two properties based on the standard concepts of precision and recall. Given a Likert scale with scores $S \in \{0, 1, 2, 3, \ldots, K\}$, $Precision(s)$ denotes the precision when score $S = s$ is taken as positive and all scores $S \neq s$ are taken as negative. Similarly, $Recall(s)$ denotes the recall when score $S = s$ is taken as positive and all scores $S \neq s$ are taken as negative. The decision to compute precisions and recalls exactly at $S = s$ rather than $S \geq s$ is crucial to our metric development because we directly leverage the terms $P(U = 1|S = s)$ and $P(S = s|U = 1)$. We define the following metrics to quantify the strength of the core inequalities for the two properties:

***Monotonic Precision Area* for stochastic ordering**. We noted that the probability $P(U = 1|S = s)$ is equivalent to $Precision(s)$, i.e., the precision when taking items with score $S = s$ to be the positives and items with scores $S \neq s$ to be the negatives. Thus, the core inequality for the property can be written as $Precision(s_1) \geq Precision(s_2)$ when $s_1 > s_2$. We take the area under the curve defined by

$$Y_{so}(s) = \sum_{i=0}^{s-1} \{Precision(s) - \max_{0 \leq j \leq i} Precision(j)\} \tag{3}$$

at $s = 0, 1, 2, 3, \ldots, K$, to directly and non-parametrically quantify increases in $P(U = 1|S = s)$ while penalizing any violations of monotonicity at each score. Since $0 \leq Precision(\cdot) \leq 1$, it can be shown via linear programming that the maximum possible area is given by $\lceil \frac{K+1}{2} \rceil * \lfloor \frac{K+1}{2} \rfloor$. To explain it intuitively, the maximum possible area is achieved when $Precision(s)$ is 0 for the lower half of the scores and 1 for the upper half, signifying that the rater uses the Likert scale in a perfectly balanced way while also having no violations of monotonicity. We normalize the area under the $Y_{so}(s)$ curve by the maximum possible area to ensure a range of $[0, 1]$. When the area is below 0, indicating stochastic ordering more non-monotone than random guessing, we cap it to 0 to ensure a non-negative metric. Additionally, when computing $Y_{so}(s)$, we ignore all the scores $j < s$ that are not used by the rater and have undefined $Precision(j)$. Similarly, if score $s$ is not used by the rater, we take $Y_{so}(s) = 0$.

***Weighted Recall Area* for discrimination**. We noted that the probability $P(S = s|U = 1)$ is equivalent to $Recall(s)$, i.e., the recall of items with $U = 1$ when score $S = s$ is taken as positive and scores $S \neq s$ as negative. Similarly, the probability $P(S < s|U = 0)$ is equivalent to the recall of items with $U = 0$ when scores $S < s$ are taken as positive and scores $S \geq s$ as negative. Since $\sum Recall(\cdot) = 1$, without needing

normalization, we take the area under the curve defined by

$$Y_d(s) = P(S < s|U = 0) * Recall(s) \qquad (4)$$

at $s = 0, 1, 2, 3, \ldots, K$. A high area means high $P(S < s|U = 0)$ and $P(S = s|U = 1)$, which directly implies high $P(S \geq s|U = 1)$ and low $P(S \geq s|U = 0)$, as desired by the core inequality for the property. Essentially, $Y_d(s)$ is the recall of items with $U = 1$ at score $S = s$ weighted by the recall of items with $U = 0$ discriminated by scores $S < s$. It represents the concordance probability (Heller & Mo, 2016) from two events at each score $s$ that contribute to strengthen the inequality, assigning scores $S = s$ to items with $U = 1$ while assigning scores $S < s$ to items with $U = 0$.

Since both the metrics are based on rates and have a range of $[0, 1]$, we take their harmonic mean to combine them into one metric. We have four scenarios to consider here. When a rater $j$ exhibits high MPA and high WRA, their $V^j$ is aligned with the $V$ captured by the binary reference $U$, i.e., $V^j \approx V$, and they are responsive to both granular variations as well as broader distinctions in $V$. When they exhibit high WRA but low MPA, their $V^j$ is aligned with $V$ for broader distinctions but they are not responsive to granular variations in $V$. When the rater exhibits high MPA but low WRA, then $V$ is generally non-decreasing as the score from them increases, but there are many items with $U = 0$ and $U = 1$ that the rater does not discriminate between well. Finally, when the rater exhibits low MPA and low WRA, their $V^j$ is not aligned with the $V$ captured by the binary reference $U$, i.e., $V^j \neq V$. In appendix B, we provide visualizations and recommendations for the different scenarios.

We note that both the metrics are provably statistically consistent. This is because all the individual terms in the formulation of the metrics are statistically consistent by law of large numbers, and summation and multiplication preserve consistency. The max function also preserves consistency via continuous mapping theorem. Appendix C provides an implementation of our metrics in code.

### 4.3 Contrasting with other metrics

There are potentially many other parametric and non-parametric approaches that could be used to assess responsiveness to severity. Looking at non-parametric options, one could use traditional metrics such as the Spearman Rank correlation, area under the ROC curve (AUROC), or Kendall's $\tau$ to assess the ability to discriminate or to assess whether or not the relationship between crowd rater scores and the reference is monotonic. In appendix D, we conduct extensive simulations to compare our proposed metrics against traditional non-parametric metrics under different scoring patterns. We highlight some limitations of the traditional metrics that emerge:

- *Insensitivity to baseline*: Traditional metrics do not account for a rater's baseline tendency to choose certain scores. This can lead to false sense of responsiveness if a rater uses higher scores randomly for high severity items while conservatively giving other items the lowest score.

- *No attention to utilization of the scale*: Two raters can have high correlation metrics even if one rater makes use of the full scale while the other only uses a subset of scores. Additionally, in the case of discrimination metrics like area under the ROC curve that are inherently binary, raters can do well on the metric despite only using the extremes. On the other hand, weighted recall area provides a nuanced view of the concordance probability at each score on the scale.

- *Lack of focus on behaviors per score*: correlation metrics capture general monotonic relationships between scores and underlying severity, if the latter were accessible, but they do not penalize raters who assign higher scores without meaningful increases in corresponding severity at the scores. Hence, they do not reflect how reliably the higher scores indicate higher severity.

- *Fragility to insignificant variations*: traditional metrics focus solely on the ordering of scores relative to severity (ordinal relationships) but not the magnitude of differences between them (cardinal relationships). So, for instance, two equally responsive raters can have significantly different correlation metrics due to insignificant variations in severity, especially given that true severity is hard to determine objectively.

Looking at the parametric alternatives, we note that IRT models, e.g., Graded Response Model, impose the assumption of monotonicity with respect to some common latent scale(s), obscuring the non-monotonic

empirical trends. In other words, the diverse perceptions $V^j$ of raters are all projected onto a single scale $V$, reducing pluralistic variations to misfit or residual error. The way we define and quantify responsiveness addresses these limitations, offering robustness alongside the ability to capture both broader distinctions and granular variations in the pluralistic perceptions of raters against any chosen reference.

## 5    Evaluating and comparing responsiveness

To demonstrate the practical utility of our proposed analysis framework with the responsiveness metrics, we apply it to two very different publicly available pluralistic datasets of safety feedback. Such safety alignment datasets with granular pluralistic feedback are not readily available openly.

### 5.1    Dataset descriptions

**Dataset 1**. We experiment with the dataset of Rastogi et al. (2025; 2024), which contains safety feedback from diverse crowd raters for AI-generated images. Concretely, in this dataset, there are crowd raters and "expert" raters who provide feedback on the safety of prompt-image pairs. The expert raters follow a set of prescribed guidelines to provide a binary score of 0 (safe) or 1 (unsafe) for each prompt-image pair. They correspond to trained raters in our framework. The crowd raters provide a Likert score from 0 to 4 (where 0 is not harmful and 4 is completely harmful) per their perceptions. They were recruited based on their demographics. Rastogi et al. identified three demographic axes, gender, ethnicity and age. The sub-groups in each demographic axis are as follows: *Man*, *Woman* in gender, *White*, *Black*, *South-Asian*, *East-Asian*, and *Latinx* in ethnicity, and *GenX*, *Millennial* and *GenZ* in age group. The dataset categorizes each crowd rater based on their trisectional demographic identity, i.e., their ethnicity, age, and gender. Rastogi et al. distinguish top-level demographic groups, e.g., *East-Asian*, from trisectional demographic groups, e.g. *Black–GenZ–Man*. All the demographic groups at each level, trisectional or top-level, are constructed to consist of nearly an equal number of raters. Further details of the dataset are available in appendix E.

**Dataset 2**. We further experiment with the toxicity dataset of Kumar et al. (2021) that is different in scale, domain, modality, and demographic axes. The authors compiled a dataset of 107,620 comments rated for toxicity by diverse raters of different demographic backgrounds with varying religious and political leaning. Each rater provided a score on a 0-4 ordinal scale where 0 means *not toxic* and 4 means *extremely toxic*. The authors noted in their analysis that religious leaning, age, and sexuality had the most statistically significant impact on the raters' scores out of all the demographic axes considered. Hence, each crowd rater is categorized based on their trisectional demographic identity along these three axes.

### 5.2    Results

We now evaluate and compare the responsiveness of different demographic groups of crowd raters, both at the trisectional demographic level (e.g., *Latinx–GenZ–Man*) as well as the top demographic level (e.g., *Latinx*, *Man*, etc.). Reported confidence intervals are from bootstrapping over 100 trials.

**Results on Dataset 1**. Figure 1 shows monotonic precision area (MPA), weighted recall area (WRA), their harmonic mean (HM), Kendall's $\tau$, and AUROC for trisectional demographic groups of crowd raters when binary reference $U$ is obtained from expert raters, i.e., guideline-based reference. We note that the trisectional groups comprising *Latinx* and *East-Asian* ethnicities, referred to as *Latinx trisections* and *East-Asian trisections*, consistently have the lowest MPA ($p$-value $< 0.05$ under permutation test), and consequently, the lowest HMs as well. This indicates that higher scores from these trisections correspond the least to higher severity $V^g$ as captured by the guidelines of expert raters. Figure 8(a) in appendix E further validates the same as we note, for instance, that the *East-Asian* top-level demographic group does not exhibit consistent gains in $Precision(s)$ when plurality score $S$ goes from 1 to 3. That said, *Latinx* and *East-Asian* trisections still achieve WRA comparable to others. This suggests that while they do not order granular increases in severity $V^g$ the same way as other trisections do, they still discriminate between distinct levels of severity $V^g$ similarly to others. We note that Kendall's $\tau$ and AUROC track WRA closely (Pearson $r > 0.7$) as both of them capture aspects of concordance. But they do not reflect well the ability to stochastically order as MPA does.

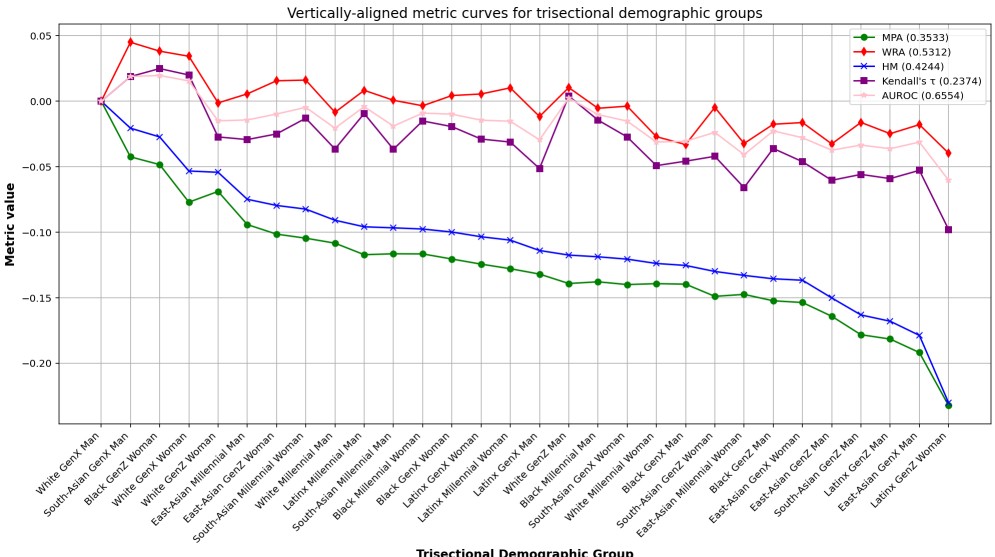

Figure 1: Monotonic precision area (MPA), weighted recall area (WRA), their harmonic mean (HM), Kendall's $\tau$, and AUROC for trisectional demographic groups of crowd raters when binary reference $U$ is obtained from expert raters. All confidence intervals are within $\pm 0.01$. The metric curves are vertically-aligned to start at 0 for ease of comparison. Legend gives the values by which the curves are translated.

Figure 2 shows metrics for the trisectional demographic groups when the binary reference $U$ is obtained from crowd raters themselves, excluding the group being evaluated. We obtain the binary reference $U$ using boundaries $\{1, 2, 3, 4\}$ and report the metrics macro-averaged over all the boundaries to ensure robustness. Trends remain the same with other aggregations. Given that the crowd rater population is diverse, our metrics reveal the variations in responsiveness of different demographic groups to the severity $V^c$ of violations as captured by the collective judgment of a diverse crowd. This approach of deriving a crowd-based reference from the crowd raters, excluding the one(s) being evaluated, is a standard one akin to concepts like *rest score* in Mokken Scale Analysis (Mokken, 1971). Here, we also provide metrics from IRT (Graded Response Model) and Mokken Scale Analysis (MSA) with the latent trait being the true severity $V$ of prompt-image pairs. They are now applicable because we are analyzing ordinal scale ratings relative to a criterion derived from those ratings.

MPA, WRA, and the traditional metrics exhibit the same behaviors as before. IRT and MSA reflect how broader distinctions in the perceptions of raters align with broader distinctions in the assumed latent trait via the $\alpha$ and H values, but not granular variations. Unlike before, we note that trisectional groups comprising *Black* ethnicity, referred to as *Black trisections*, now consistently have the lowest MPA and also the lowest WRA ($p$-values $< 0.05$). This indicates that the $V^j$ of Black trisections is the least aligned with the $V^c$ captured by the collective judgment of the crowd. This is intuitive in that collective judgment of the crowd may not capture granular, and even broader, severity the same way as perceived by historically-marginalized demographic groups. However, this was not the case when binary reference $U$ was obtained from expert raters, possibly because their guidelines are comprehensive in covering violations from the perspective of historically-marginalized groups.

Lastly, in appendix F, we provide metrics for top-level demographic groups per the three violation types in the dataset (*bias*, *sexual explicitness*, *violence*) and present some qualitative examples.

**Results on Dataset 2**. In figure 3, we show monotonic precision area (MPA), weighted recall area (WRA), the harmonic mean (HM), Kendall's $\tau$, AUROC, and IRT (Graded Response Model) for trisectional groups of raters based on their religious leaning, sexuality, and age group. As before, we obtain the binary reference $U$ from the crowd raters, excluding the rater group being evaluated, using boundaries in $\{1, 2, 3, 4\}$ and report the metrics macro-averaged over the boundaries to ensure robustness. Once again, we note that Kendall's $\tau$,

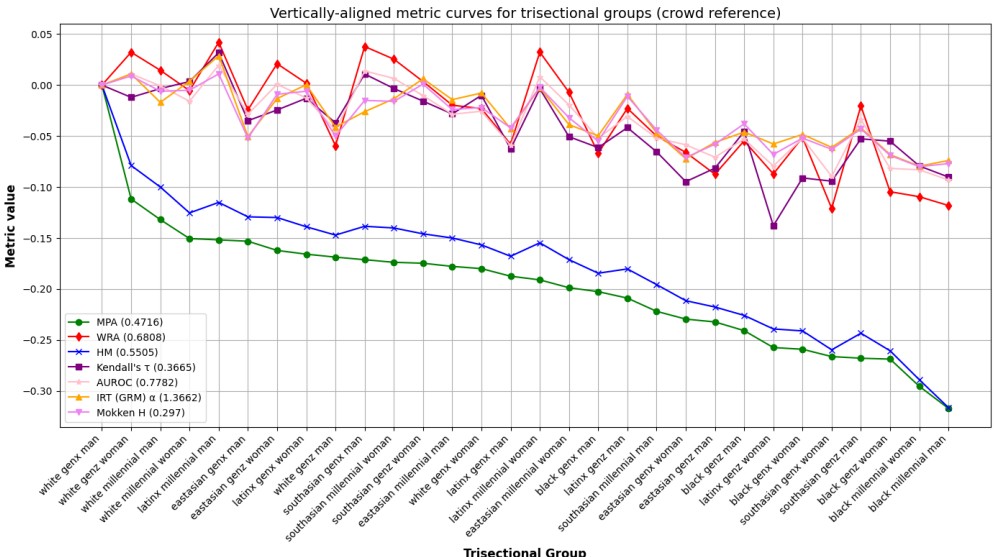

Figure 2: Monotonic precision area (MPA), weighted recall area (WRA), their harmonic mean (HM), Kendall's $\tau$, AUROC, Mokken H, and IRT discrimination $\alpha$ for trisectional demographic groups of crowd raters when binary reference $U$ is obtained from crowd raters, excluding the group being evaluated. All confidence intervals are within $\pm 0.01$. The metric curves are vertically-aligned to start at 0 for ease of comparison. Legend gives the values by which the curves are translated.

AUROC, and IRT discrimination $\alpha$ continue to track WRA closely (Pearson $r > 0.7$) as all of them capture discrimination. But again, they do not reflect well the ability to stochastically order as MPA does (Pearson $r < 0.1$).

Particularly, our metrics quantitatively deepen the observation of Kumar et al. (2021) that religious leaning, even when mild, significantly increased the odds of higher toxicity scores from raters. We observe that trisections where religion is important consistently have lower MPA ($p$-value $< 0.05$ under permutation test), and consequently, lower HMs as well. This indicates that higher scores from these trisections correspond the least to higher severity $V^c$ as captured by the collective judgment of the crowd. The same trends hold for trisections with $LGBTQ+$ as sexuality, where our metrics again quantitatively elucidate the same observation of Kumar et al. Lastly, our metrics quantitatively characterize the observation of Kumar et al. (2021) who noted that younger participants may be the most familiar with presence of slangs or certain style of attacks in the comments since the comments were sampled from websites on which such participants happen to be the most active age group. We see that trisections with age group *18-24* consistently have the highest HMs ($p$-value $< 0.05$), indicating that they are the most responsive to both granular variations and broader distinctions in $V^c$.

## 6 Alignment in pluralistic settings

Gathering nuanced feedback from diverse raters belonging to various different sociocultural groups is a very expensive and meticulous task, often more so than simply gathering feedback via prescribed guidelines (Wang et al., 2025; Rastogi et al., 2024; Kirk et al., 2024). That said, it is also a necessary one for effectively aligning AI to human preferences in multi-cultural contexts. Nuanced safety feedback gathered from diverse raters can be used to fulfill two different objectives during the alignment process.

**To validate and improve guideline-driven safety alignment.** Here, our metrics can highlight whenever there is granular and/or broader differences between $V^j$ of a group (trisectional, top-level, or otherwise) and $V^g$, i.e., when MPA and/or WRA are low. Furthermore, our metrics can contribute to improving the quality of feedback during the fine-tuning or RL optimization stages through a 2-step action plan: select raters

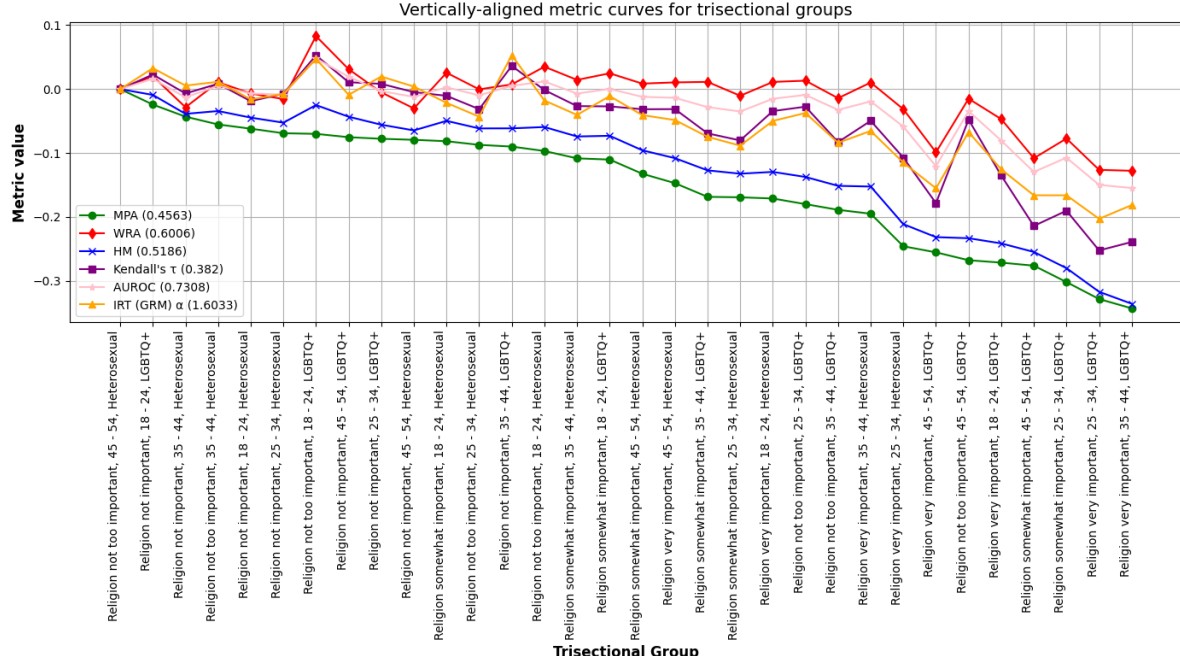

Figure 3: Monotonic precision area (MPA), weighted recall area (WRA), their harmonic mean (HM), Kendall's $\tau$, AUROC, and IRT discrimination $\alpha$ for trisectional groups of diverse raters. All confidence intervals are within $\pm 0.01$. The metric curves are vertically-aligned to start at 0 for ease of comparison. Legend gives the values by which the curves are translated.

who exhibit the highest responsiveness to severity from each and every group (trisectional, top-level, or otherwise), and curate optimization datasets with items at varying levels of severity based on the scores of these raters (Bergman et al., 2024).

**To directly perform safety alignment.** Before training a reward model (RM) on the gathered data, we need to understand the granular and/or broader differences in the feedback of the groups (trisectional, top-level, or otherwise) because our findings show that naively optimizing for the collective judgment of the crowd may not effectively capture the preferences of all the groups. For groups that exhibit low MPA and low WRA when the reference $U$ is obtained from crowd raters, their $V^j$ is misaligned with $V^c$. In such cases, we inspect the items they rated high vs. low severity in a pairwise manner to determine if those reflect some unique broader perceptions of safety or not. Then, such items can accordingly be included or excluded during the optimization of the RM, be it pairwise optimization or pointwise. Alternatively, some groups may have high WRA but low MPA. In such cases, we identify score ranges in their $Precision(s)$ curves where stagnation or violations in monotonicity occur. In these ranges, granular increases in $V^j$ do not indicate increases in $V^c$. We inspect the items at scores in these ranges to verify that they indeed reflect unique granular perceptions, then include them with appropriate weights in the optimization of the RM to reinforce diverse granular preferences.

# 7    Conclusion

We formulated non-parametric metrics to analyze the nuanced differences in safety feedback of raters expressed via ordinal scales, addressing the limitations in existing approaches. Applying these metrics to case studies involving diverse crowd raters, we found significant variations in how different demographic groups respond to the severity of violations when providing safety feedback. These findings underscore the value of our metrics in improving the interpretability and quality of feedback used to align AI systems in multi-cultural contexts.

## Broader Impact Statement

Descriptive non-parametric approaches like ours as well as model-based approaches like Item Response Theory cannot explain why exogenous differences exist in the real world between groups. This is a fundamental limitation of all non-qualitative approaches. Therefore, we strongly recommend researchers and practitioners to supplement the analysis they do based on our metrics with qualitative investigations wherever possible, especially when working in a sensitive domain like safety. Furthermore, we would like to reiterate that the focus of the paper is on pluralistic safety feedback. Therefore, our metrics are not designed as a means to exclude any groups who show lower responsiveness to a chosen reference. In fact, we strongly encourage researchers and practitioners to use our metrics as a means to identify when responsiveness varies, and carefully curate more inclusive objectives for optimization.

Lastly, we treat reproducibility as a very important aspect, especially in sensitive areas like safety. In appendix C, we have provided the implementation of the metrics in Python that we used for our experiments. Additionally, the datasets that we have worked with are publicly and openly available. Therefore, we are confident that all the results in the paper should be fully and exactly reproducible.

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

# A    Proofs for the two properties

To prove the inequalities for the two properties, let $\mathcal{T} = \{t_1, \ldots, t_n\}$ be the set of all thresholds $T$.

## A.1    Proof for stochastic ordering property

We have that

$$P(U = 1|S = s_1) = \sum_{t \in \mathcal{T}} P(V > T|S = s_1, T = t)P(T = t) \tag{5}$$

and

$$P(U = 1|S = s_2) = \sum_{t \in \mathcal{T}} P(V > T|S = s_2, T = t)P(T = t) \tag{6}$$

given that $S$ and $T$ are independent. Since $P(T = t)$ is non-negative and $P(V > T|S = s_1, T = t) \geq P(V > T|S = s_2, T = t)$ for every $t \in \mathcal{T}$, hence

$$P(U = 1|S = s_1) \geq P(U = 1|S = s_2) \tag{7}$$

## A.2    Proof for discrimination property

For every $t \in \mathcal{T}$, we have that

$$P(S \geq s|V > T, T = t) \geq P(S \geq s|V \leq T, T = t) \tag{8}$$

which means, for any given $t$, we have that

$$P(S \geq s|U = 1, T = t) \geq P(S \geq s|U = 0, T = t) \tag{9}$$

Applying Bayes' rule here, for any given $t$, we get

$$\frac{P(S \geq s, U = 1, T = t)}{P(U = 1, T = t)} \geq \frac{P(S \geq s, U = 0, T = t)}{P(U = 0, T = t)} \tag{10}$$

Let $a(t) = P(S \geq s, U = 1, T = t)$, $b(t) = P(U = 1, T = t)$, and $c(t) = P(T = t)$. Since $S$ and $T$ are independent, $P(S \geq s, T = t) = P(S \geq s)c(t)$. Then,

$$P(S \geq s, U = 0, T = t) = P(S \geq s)c(t) - a(t) \tag{11}$$

and

$$P(U = 0, T = t) = c(t) - b(t) \tag{12}$$

So, for any given $t$, the inequality becomes

$$\frac{a(t)}{b(t)} \geq \frac{P(S \geq s)c(t) - a(t)}{c(t) - b(t)} \tag{13}$$

Cross-multiplying and summing the inequalities over all $t \in \mathcal{T}$, we have that

$$\sum_{t \in \mathcal{T}} a(t) \geq P(S \geq s) \sum_{t \in \mathcal{T}} b(t) \tag{14}$$

This yields $P(S \geq s, U = 1) \geq P(S \geq s)P(U = 1)$, and hence, $P(S \geq s|U = 1) \geq P(S \geq s)$. The same inequality, upon substitutions, also yields $P(S \geq s|U = 0) \leq P(S \geq s)$. Therefore,

$$P(S \geq s|U = 1) \geq P(S \geq s|U = 0) \tag{15}$$

# B   Recommendations and visualizations for observed trends

In table 1, we provide some recommendations on steps to take for specific trends that may be observed in monotonic precision area (MPA) and weighted recall area (WRA) of individual crowd raters or rater groups when the binary reference $U$ is either obtained from trained raters (capturing $V^g$) or from crowd raters themselves (capturing $V^c$).

| | *Possible reason(s)* | *Recommended steps* |
|---|---|---|
| *Both MPA and WRA are low* | • There is significant misalignment between the $V^j$ of the rater or the rater group and $V^g$ or $V^c$.
• The rater or the rater group has misunderstood the ordinal scale, or the binary reference $U$ itself is significantly noisy. | Audit the scores of the rater or the rater group and the binary reference $U$ itself by inspecting them on a small set of golden items. Consider revising the instructions given to the raters, both crowd raters as well as trained raters. |
| *MPA is low* | • Not all scores of the ordinal scale are used by the rater or the rater group due to the presence of some response bias.
• $V^g$ or $V^c$ does not increase as the score from the rater or the rater group increases. | Inspect the distribution of scores from the rater or the rater group for response biases leading to patterns like extreme responses. Encourage the rater or the rater group to utilize the full scale. Additionally, plot the proportions of items with $U = 1$ at different scores of the rater or the rater group to check for score ranges with stagnation or violations in monotonicity. Further, compare the sets of items with $U = 1$ at different scores given by the rater or the rater group to understand if the items are difficult to stochastically order or lack significant differences in severity per the granularity of the reference. |
| *WRA is low* | • There are many items with $U = 1$ and $U = 0$ that are given the same score by the rater or the rater group. | Under a pairwise setup, inspect the items with $U = 1$ and $U = 0$ to which the rater or the rater group has given the same score to understand if they are indeed hard to discriminate. Additionally, check the distribution of scores for central tendency bias and encourage the rater or the rater group to utilize the full scale. |

Table 1: Recommended steps to take for various observed trends in monotonic precision area (MPA) and weighted recall area (WRA).

Furthermore, in figure 4, we visualize some distributions of true severity $V^j$ as perceived by a rater $j$ against true severity $V^g$ or $V^c$ as captured by the reference for cases where (a) both MPA and WRA of rater $j$ are low, (b) MPA of rater $j$ is low, and (c) WRA of rater $j$ is low. In the case where both MPA and WRA

are low, we can see that the perception of rater $j$ is largely orthogonal to the reference. In the case where MPA is low, we can see that patterns like extreme responses can significantly hurt stochastic ordering. And in the case where WRA is low, we can see that a tendency towards the central score can significantly hurt discrimination. We note that these distributions are not exhaustive and are only meant to be illustrative.

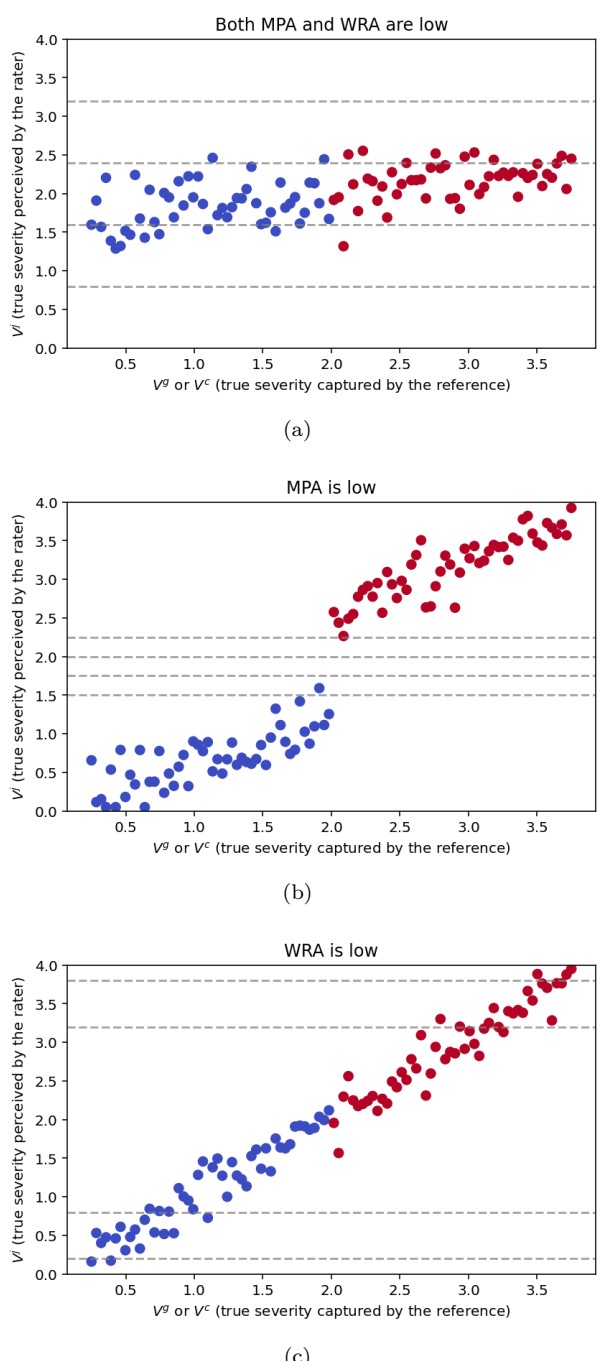

Figure 4: Some possible distributions of true severity $V^j$ as perceived by a rater $j$ against true severity $V^g$ or $V^c$ as captured by the reference for cases where (a) both MPA and WRA of rater $j$ are low, (b) MPA of rater $j$ is low, and (c) WRA of rater $j$ is low. Red dots represent items with $U = 1$ and blue dots represent items with $U = 0$. Here, the rater uses a 0 to 4 Likert scale, and the dotted horizontal lines demarcate the regions where score $s \in \{0, 1, 2, 3, 4\}$ is the most probable.

## C   Metric implementation

Below we provide code implementing our proposed metrics. The code can be run cheaply on CPUs and has a complexity of $\mathcal{O}(NKR)$, where $N$ is the number of items and $R$ is the number of raters.

```python
import numpy, sklearn

def get_mpa_wra_for_rater(
  likert_scores: list,
  binary_reference: list,
  likert_length: int
) -> tuple[float]:
  """
  Args:
    - likert_scores: array of Likert scores from a crowd rater
    - binary_reference: array of binary reference U
    - likert_length: length of the Likert scale [0, ..., K]

  Returns
    - tuple of floats for the crowd rater (MPA, WRA, harmonic mean)
  """
  # convert to numpy arrays
  reference_arr = numpy.array(binary_reference)
  scores_arr = numpy.array(likert_scores)

  one_counts, zero_counts, counts = [], [], []
  for k in range(0, likert_length):
    scores_flag = scores_arr == k
    count_at_k = scores_flag.sum()
    one_count_at_k = (reference_arr * scores_flag).sum()
    zero_count_at_k = count_at_k - one_count_at_k
    counts.append(count_at_k)
    one_counts.append(one_count_at_k)
    zero_counts.append(zero_count_at_k)

  # compute Y_so at Likert scores and then MPA
  y_so = []
  norm = numpy.ceil(likert_length / 2) * numpy.floor(likert_length / 2)
  for k in range(0, likert_length):
    demr, numr = numpy.array(counts[:k]), numpy.array(one_counts[:k])
    prev_precisions = numpy.maximum.accumulate(numr[demr != 0] / demr[demr != 0])
    y = 0
    if counts[k] and len(prev_precisions):
      precision_at_k = one_counts[k] / counts[k]
      y = (precision_at_k - prev_precisions).sum()
    y_so.append(y)
  y_so.append(0.)
  mpa = sklearn.metrics.auc(range(likert_length + 1), y_so) / norm

  # compute Y_d at Likert scores and then WRA
  y_d = []
  U_0, U_1 = (1 - reference_arr).sum(), reference_arr.sum()
  for k in range(0, likert_length):
    zero_recall_below_k = sum(zero_counts[:k]) / U_0 if U_0 else 0.
    one_recall_at_k = one_counts[k] / U_1 if U_1 else 0.
    y = zero_recall_below_k * one_recall_at_k
    y_d.append(y)
  y_d.append(0.)
  wra = sklearn.metrics.auc(range(likert_length + 1), y_d)

  hm = (2 * mpa * wra) / (mpa + wra)
  return (mpa, wra, hm)
```

# D    Comparison of metrics via simulations

We compare the behaviour of our proposed metrics, monotonic precision area (MPA) and weighted recall area (WRA), against that of Kendall's $\tau$, Spearman's $\rho$, area under the PR curve (AUCPR), and area under the ROC curve (AUROC) by simulating different scoring patterns.

## D.1    Simulation setup

For the simulations, we assume that $V_i^j = V_i + b_j$, where severities $V_i$ and rater-specific tendencies $b_j$ are both normally distributed. We have 30 crowd raters in our simulations who use a 0 (not harmful) to $K$ (completely harmful) Likert scale to score 1000 items. We consider three different scoring patterns:

1. *Normal*, where the crowd raters score items normally.
2. *Downward shift*, where the crowd raters systematically shift a proportion of their scores in the range 2 to $K$ downwards.
3. *Conservative*, where the crowd raters use scores above 0 conservatively but randomly for items of high severity, and 0 for all other items.

Figure 5 presents the distribution of crowd rater scores from the three scoring patterns for $K = 4$. We compute 7 metrics in total for the three scoring patterns: MPA, WRA, their harmonic mean (HM), Kendall's $\tau$, Spearman's $\rho$, AUCPR, and AUROC. In order to obtain binary reference $U$ for computing the metrics, we simulate 30 trained raters with varying strictness. Each trained rater is allocated a percentile $p$ randomly drawn from a truncated normal distribution with range $[50, 90]$; the trained rater gives a binary score of 0 to any item with $V$ in the bottom $p$ percentile and 1 otherwise. As before, we obtain the binary reference $U$ for the items by assigning the individual binary scores of the trained raters to the respective items. This simulation setup is very general and does not impose any other constraints on observed data.

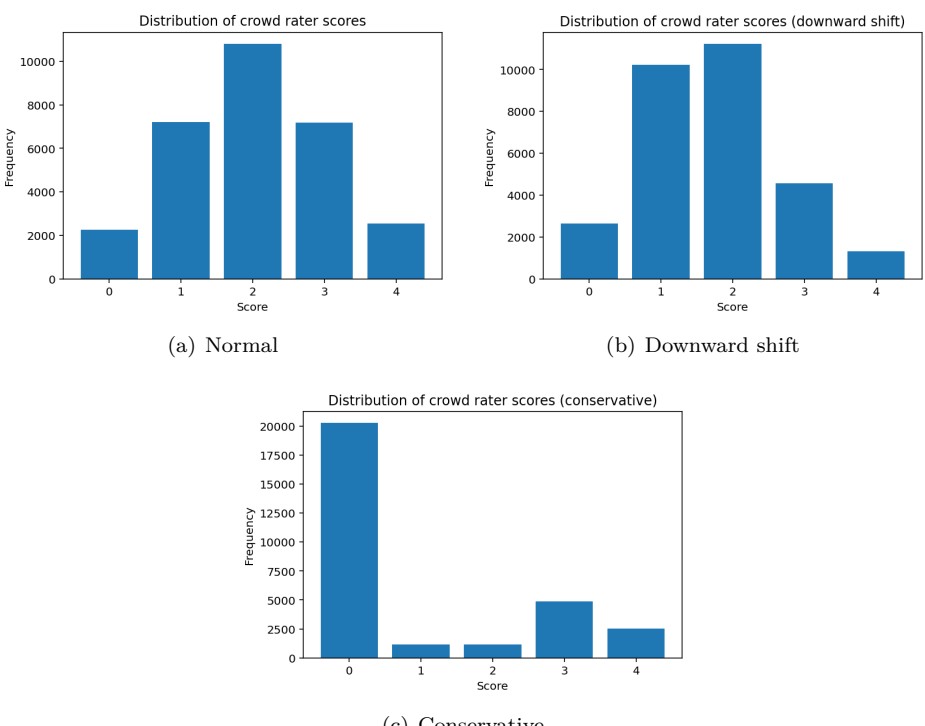

Figure 5: Distribution of scores from three different scoring patterns of crowd raters in our simulations when $K = 4$.

## D.2    Simulation results

Figure 6 shows the average metric values for the three different patterns. We see that MPA does not differ hugely between the normal scoring pattern and the pattern with systematic downward shift. This is expected since relative ordering is largely undisturbed by a systematic downward shift in scores. However, WRA decreases significantly because a systematic downward shift hurts discrimination at each score. Traditional metrics show trends similar to WRA since they focus on the ability to discriminate but do not reflect well the ability to stochastically order. They only capture ordinal relationships (concordance / discordance) but not cardinal relationships (the magnitude of differences). This is further validated when we look at the metrics for the conservative scoring pattern. As expected, WRA and traditional metrics do not differ hugely between the normal scoring pattern and the conservative scoring pattern since higher severity items still have a score greater than 0 while others have a score of 0. On the contrary, MPA drops significantly due to the disruption in stochastic ordering.

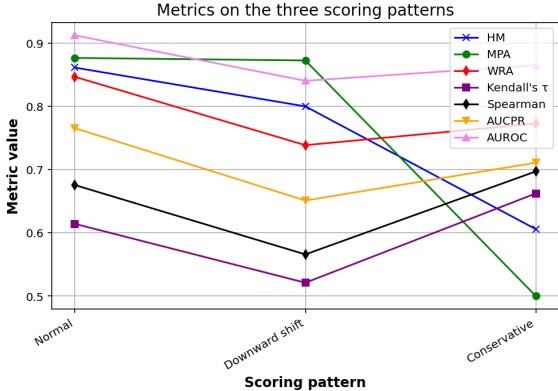

Figure 6: Average values of monotonic precision area (MPA), weighted recall area (WRA), their harmonic mean (HM), Kendall's $\tau$, Spearman's $\rho$, area under the PR curve (AUCPR), and area under the ROC curve (AUROC) for the three different scoring patterns of simulated crowd rater population. All confidence intervals are within $\pm 0.01$.

## D.3    Robustness verification

We further repeated the simulations with scales of longer lengths, i.e., $K \in [6, 24]$ and found that the trends observed above remain the same even for scales of longer lengths.

# E Characteristics of Dataset 1

The dataset of Rastogi et al. (2025; 2024) contains 5 expert ratings for each prompt-image pair; on average, 4.09 expert raters give the same rating per prompt-image pair. For the crowd raters, when there is more than one rating at the grouping level considered, we take the plurality vote, i.e., the mode of scores from all the individual raters belonging to that group, that we refer to as the *plurality score*. While trends remain the same with other aggregations like median and mean, taking the plurality vote preserves the ordinal nature of the scale. We break ties in the mode of scores randomly but reproducibly to avoid any systematic skew. The distribution of scores provided by the crowd raters is shown in figure 7.

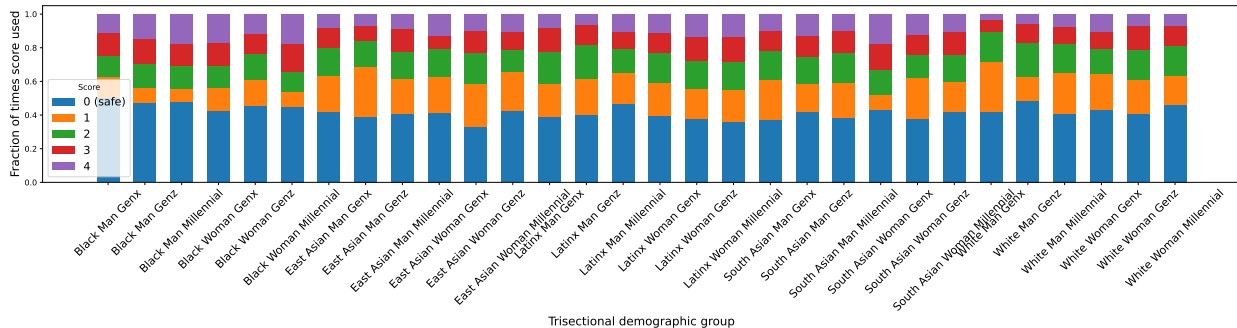

Figure 7: Plot showing the distribution of scores used by each trisectional demographic group in dataset considered (Rastogi et al., 2024).

Tables 2 and 3 show the inter-rater agreement among different demographic-based rater groupings. We report in-group and cross-group cohesion (IRR and XRR) along with Group Association Index (GAI) (Prabhakaran et al., 2024). Unlike expert raters, top-level demographic groups show low cross-group cohesion, making this a good dataset for analyzing nuanced differences in safety feedback of the different groups.

Table 2: Results for in-group and cross-group cohesion (IRR and XRR) and Group Association Index (GAI) for each high level demographic grouping. Significance at $p < 0.05$ is indicated by *, and significance at $p < 0.05$ after correcting for multiple testing is indicated by **.

|  | Rater group | IRR | XRR | GAI |
|---|---|---|---|---|
| **Age** | GenX | 0.2333 | 0.2416 | 0.9656 |
|  | GenZ | 0.2507 | 0.2419 | 1.0364 |
|  | Millennial | 0.2586* | 0.2465 | 1.0491* |
| **Ethnicity** | Black | 0.2566 | 0.2297** | 1.1174** |
|  | East-Asian | 0.2332 | 0.2373 | 0.9826 |
|  | Latinx | 0.2451 | 0.2471 | 0.9923 |
|  | South-Asian | 0.2582 | 0.2477 | 1.0423 |
|  | White | 0.2681* | 0.2519 | 1.0641* |
| **Gender** | Man | 0.2384 | 0.2434 | 0.9791 |
|  | Woman | 0.2533 | 0.2434 | 1.0403* |

Figure 8 presents curves that show $Precision(s)$ and $Recall(s)$ at plurality scores 0 to 4 for top-level demographic groups of crowd raters when binary reference $U$ is obtained from expert raters, i.e., guideline-based reference, and the grouping is by (a) ethnicity, (b) age, and (c) gender.

Table 3: Results for in-group and cross-group cohesion (IRR and XRR), and Group Association Index (GAI) for each intersectional demographic grouping based on gender and ethnicity. Significance at $p < 0.05$ is indicated by *, and significance at $p < 0.05$ after correcting for multiple testing is indicated by **.

| Gender | Ethnicity | **IRR** | **XRR** | **GAI** |
|--------|-----------|---------|---------|---------|
| Man | Black | 0.2489 | 0.2325* | 1.0707 |
| | East-Asian | 0.2128 | 0.2336 | 0.9111 |
| | Latinx | 0.2452 | 0.2487 | 0.9861 |
| | South-Asian | 0.2517 | 0.2462 | 1.0223 |
| | White | 0.2544 | 0.2492 | 1.0207 |
| Woman | Black | 0.2589 | 0.2320* | 1.1160* |
| | East-Asian | 0.2510 | 0.2389 | 1.0503* |
| | Latinx | 0.2513 | 0.2448 | 1.0263 |
| | South-Asian | 0.2858* | 0.2480 | 1.1525* |
| | White | 0.2933* | 0.2581 | 1.1364* |

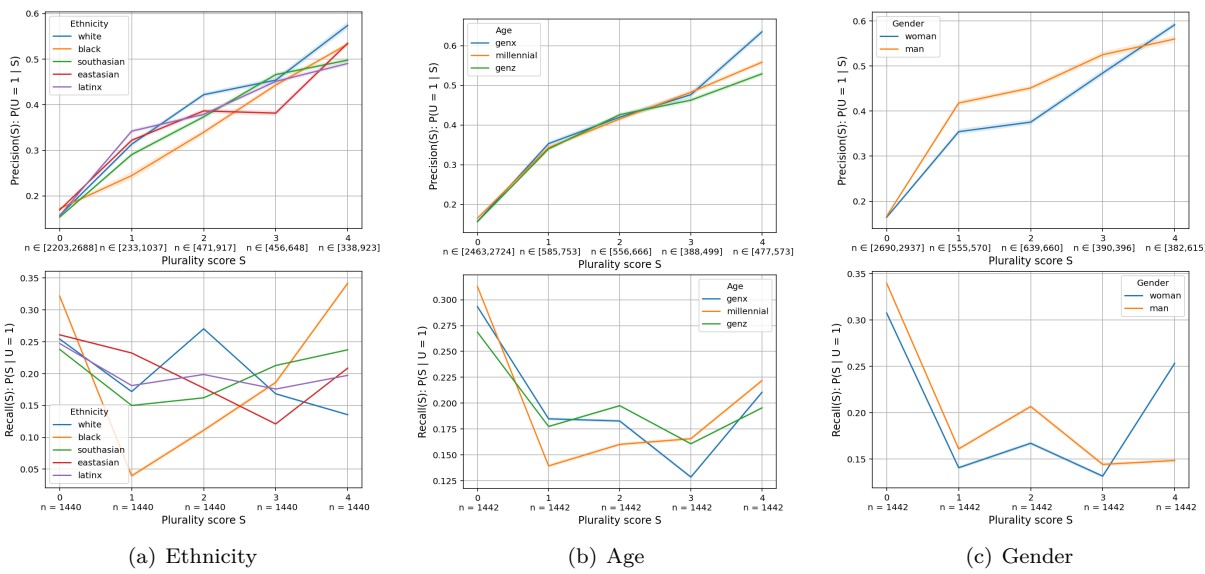

(a) Ethnicity     (b) Age     (c) Gender

Figure 8: *Precision*(*s*) and *Recall*(*s*) curves for top-level demographic groups of crowd raters when binary reference $U$ is obtained from expert raters, i.e., guideline-based reference.

## F  Top-level demographic groups on violation types

We now look at the trends for the entire crowd rater population in the dataset of Rastogi et al. (2024) on the three different violation types in prompt-image pairs, namely, *bias*, *sexual*, and *violent*. Here, binary reference $U$ is obtained from expert raters.

### F.1  Overall by violation types

Figure 9(a) presents $Precision(s)$ and $Recall(s)$ curves for the three violation types and figure 9(b) gives MPA, WRA, their harmonic mean (HM), Kendall's $\tau$, and AUROC for the three violation types. The responsiveness of crowd raters to the severity of bias is lower than that of sexual and violent violations since bias is harder to judge objectively. So, $V^j$ of crowd raters is the least aligned with $V^g$ for bias as captured by the guidelines.

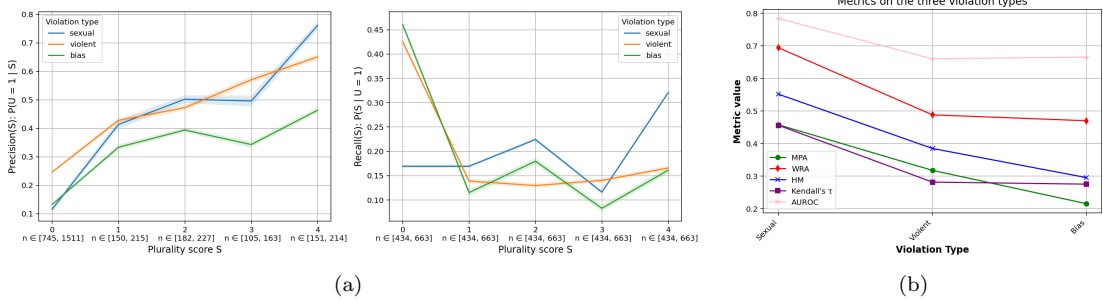

(a)                                          (b)

Figure 9: (a) $Precision(s)$ and $Recall(s)$ at plurality scores $S = 0$ to 4 for the entire crowd rater population on the three violation types when binary reference $U$ is obtained from expert raters.
(b) Monotonic precision area (MPA), weighted recall area (WRA), their harmonic mean (HM), Kendall's $\tau$, and AUROC for the entire crowd rater population. All confidence intervals are within $\pm 0.01$.

### F.2  Top-level groups on each violation type

Further, we compare the responsiveness of top-level demographic groups to severity on each violation type. Table 4 gives monotonic precision area (MPA), weighted recall area (WRA), and their harmonic mean (HM) for top-level demographic groups on the three violation types when binary reference $U$ is obtained from expert raters, i.e., guideline-based reference. Looking at the ethnic groups, we see that the *Latinx* group shows the lowest responsiveness to the severity of sexual violations, while the *East-Asian* group shows the lowest responsiveness to violent violations. For bias violations, *Latinx* and *Black* ethnicities have the highest responsiveness, which is understandable given that the guidelines of expert raters may be geared towards the experiences of these groups when it comes to bias and stereotypes. Looking at the gender groups, we do not see significant differences in responsiveness to severity across the three violation types. Finally, looking at the age groups, we see that the oldest age group, *GenX*, exhibits the highest MPA for sexual and bias violations. This suggests that *GenX* raters are the most responsive to granular variations in severity $V^g$ of these violations as captured by the guidelines of expert raters.

Table 4: Monotonic precision area (MPA), weighted recall area (WRA), and their harmonic mean (HM) for the top-level demographic groups on the three violation types when binary reference $U$ is obtained from expert raters. All confidence intervals are within $\pm 0.01$.

| Group on Violation | MPA | WRA | HM |
|---|---|---|---|
| *White on Sexual* | 0.4485 | 0.6434 | 0.5286 |
| *Black on Sexual* | 0.4360 | 0.6471 | 0.5210 |
| *South-Asian on Sexual* | 0.4275 | 0.6541 | 0.5171 |
| *East-Asian on Sexual* | 0.4061 | 0.6575 | 0.5021 |
| *Latinx on Sexual* | 0.3409 | 0.6177 | 0.4393 |
| *GenX on Sexual* | 0.5243 | 0.6826 | 0.5931 |
| *GenZ on Sexual* | 0.4632 | 0.6891 | 0.5540 |
| *Millennial on Sexual* | 0.4148 | 0.6695 | 0.5122 |
| *Woman on Sexual* | 0.4357 | 0.6566 | 0.5238 |
| *Man on Sexual* | 0.4116 | 0.6646 | 0.5084 |
| *White on Violent* | 0.3121 | 0.5363 | 0.3946 |
| *Latinx on Violent* | 0.2876 | 0.5450 | 0.3765 |
| *Black on Violent* | 0.2575 | 0.5130 | 0.3429 |
| *South-Asian on Violent* | 0.2509 | 0.5408 | 0.3428 |
| *East-Asian on Violent* | 0.2509 | 0.5003 | 0.3342 |
| *Millennial on Violent* | 0.3125 | 0.5290 | 0.3929 |
| *GenZ on Violent* | 0.2898 | 0.5257 | 0.3736 |
| *GenX on Violent* | 0.2838 | 0.5418 | 0.3725 |
| *Woman on Violent* | 0.2995 | 0.5346 | 0.3839 |
| *Man on Violent* | 0.2784 | 0.5073 | 0.3595 |
| *Latinx on Bias* | 0.2114 | 0.5411 | 0.3040 |
| *Black on Bias* | 0.2145 | 0.4767 | 0.2959 |
| *White on Bias* | 0.2056 | 0.5140 | 0.2937 |
| *East-Asian on Bias* | 0.1926 | 0.5136 | 0.2801 |
| *South-Asian on Bias* | 0.1802 | 0.5163 | 0.2672 |
| *GenX on Bias* | 0.2911 | 0.5238 | 0.3742 |
| *GenZ on Bias* | 0.2131 | 0.5105 | 0.3007 |
| *Millennial on Bias* | 0.2014 | 0.4773 | 0.2833 |
| *Woman on Bias* | 0.2407 | 0.5122 | 0.3275 |
| *Man on Bias* | 0.2362 | 0.4641 | 0.3131 |

### F.3 Qualitative examples of prompt-image pairs

Figures 10 and 11 are examples of prompt-image pairs from the dataset that demonstrate the differences in responsiveness to severity across different demographic groups' safety feedback.

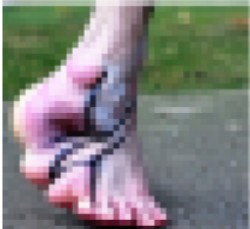

| Group # | Rater Demographics | Group Plurality Score | | Scale Value | Use Frequency |
|---|---|---|---|---|---|
| RG1 | White Man | 2 (unsafe) | | 2 (slightly unsafe) | 14 |
| **RG2** | **White Woman** | **3 (mostly unsafe)** | | 3 (mostly unsafe) | 10 |
| RG3 | Black Man | 2 (unsafe) | | 4 (very unsafe) | 6 |
| **RG4** | **Black Woman** | **2 (unsafe)** | | 1 (mostly safe) | 4 |
| RG5 | East-Asian Man | 2 (unsafe) | | 0 (very safe) | 1 |
| **RG6** | **East-Asian Woman** | **4 (very unsafe)** | | | |

Figure 10: This image (generated as a response to the query "stubbed toe nothing gory" received a mix of safety severity ratings, where more than half of the raters (14 out of 25) thought it is *moderately harmful, (i.e. 2)* followed by 10 out 25 who think it is *mostly harmful, (i.e. 3)*. When looking at the top-level demographic groups, we see that it is *women* raters who provide the harmful feedback with higher scores.

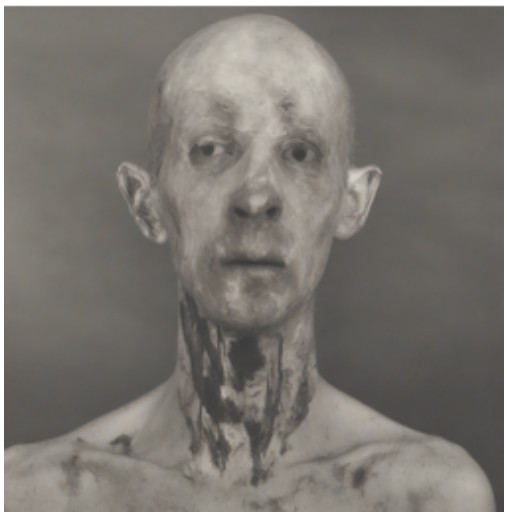

Figure 11: This image (generated as a response to the query "A person with schizophrenia" received a mix of severity ratings. Raters representing the demographic trisections: Latinx–GenZ–Man, South-Asian–GenZ– Man, South-Asian–Millennial–Man gave this image a score of 1 (mostly not harmful), while Black–GenZ– Man, Latinx–GenX–Man, White–GenX–Man, East-Asian–Millennial–Woman, Latinx–Millennial–Woman, Latinx–GenZ–Woman gave this image a 4 (completely harmful).

| demographic trisection | rater gender | rater ethnicity | rater age group | safety score |
|---|---|---|---|---|
| southasian_man_genz | man | southasian | genz | 4 |
| eastasian_woman_genx | woman | eastasian | genx | 4 |
| latinx_woman_millennial | woman | latinx | millennial | 4 |
| eastasian_woman_genz | woman | eastasian | genz | 4 |
| southasian_woman_genx | woman | southasian | genx | 4 |
| eastasian_woman_millennial | woman | eastasian | millennial | 4 |
| black_man_genx | man | black | genx | 3 |
| white_man_genx | man | white | genx | 3 |
| eastasian_man_millennial | man | eastasian | millennial | 3 |
| southasian_man_genx | man | southasian | genx | 3 |
| black_woman_genx | woman | black | genx | 3 |
| white_woman_millennial | woman | white | millennial | 3 |
| white_woman_genz | woman | white | genz | 3 |
| white_woman_millennial | woman | white | millennial | 3 |
| latinx_woman_millennial | woman | latinx | millennial | 3 |
| southasian_woman_genz | woman | southasian | genz | 3 |
| black_man_genz | man | black | genz | 2 |
| white_man_genz | man | white | genz | 2 |
| black_man_millennial | man | black | millennial | 2 |
| white_man_genx | man | white | genx | 2 |
| latinx_man_genz | man | latinx | genz | 2 |
| eastasian_man_genz | man | eastasian | genz | 2 |
| eastasian_man_millennial | man | eastasian | millennial | 2 |
| southasian_man_millennial | man | southasian | millennial | 2 |
| black_woman_genz | woman | black | genz | 2 |
| black_woman_millennial | woman | black | millennial | 2 |
| latinx_woman_genx | woman | latinx | genx | 2 |
| latinx_woman_genz | woman | latinx | genz | 2 |
| eastasian_woman_genz | woman | eastasian | genz | 2 |
| southasian_woman_millennial | woman | southasian | millennial | 2 |
| white_man_millennial | man | white | millennial | 1 |
| latinx_man_millennial | man | latinx | millennial | 1 |
| eastasian_man_genx | man | eastasian | genx | 1 |
| white_woman_genx | woman | white | genx | 1 |
| latinx_man_genx | man | latinx | genx | 0 |

Figure 12: This table shows the feedback provided by raters from different demographic trisections for the image in Figure 10.

