# OpenReview forum: "Decoding Safety Feedback from Diverse Raters: A Data-driven Lens on Responsiveness to Severity"
_TMLR — Accepted by TMLR_

### Review · Reviewer_MJ7R · 2025-10-18

**Summary Of Contributions:**

This paper considers the problem of measuring how responsive raters are to safety violations in their ratings of items. Raters could either be crowd raters or professional raters. This sort of work is relevant for AI alignment, as we will often use raters to rate how “safe” AI responses are and use these rating for training. The paper introduces two metrics to measure this responsiveness for individual raters. Specifically, under mild assumptions about the relation between the true severity of violation for an item, the perceived severity and the score given to it by a rater, the paper introduces two properties to capture mathematically how responsive raters are to different severities of violation. These are the raters’ ability to order severity (assigning higher score to items whose violation is more severe) and ability to discriminate between different levels of severity (assigning different scores to items with different levels of severity). The paper then turns these into metrics and shows how they can be calculated from ratings data. Experimentally it shows that on real data these metrics differ between different demographic groups, and that these difference align with conclusions from prior work.

**Audience:**

Yes

**Audience Explanation:**

Broadly I do think that some people working on AI alignment and LLMs in particular might find this paper interesting. I do nevertheless think that the fit of this paper to TMLR is its weakest point. Fundamentally the paper introduces metrics to understand how people use ratings, allowing for analysis of rating behaviours across demographic groups. To me this is not yet readily usable for ML researchers. The paper does provide some guidance on how these metrics can be used (in section 6 and in appendix B), and for that reason I think the paper is still a good candidate for TMLR. But in my view it would have been a better fit, and a much stronger paper in general, if the authors had put this guidance into practice and had shown how and by how much these metrics can actually improve alignment.

**Broader Impact Concerns:**

Concerns are properly addressed in the broader impact statement.

**Claims And Evidence:**

Yes

**Claims Explanation:**

The modeling assumptions are mild and as far as I can tell the derivation of the metrics from these assumptions is correct. The paper’s methodology is described clearly and accurately and — apart from one minor point (see below) — was easy to follow. In the experiments the authors verify that the proposed metrics reflect conclusions from earlier work around the responsiveness of various demographic groups. Reproducibility of the experiments also appears excellent, with plenty of details and additional results in the appendices.

**Requested Changes:**

major points (critical for acceptance):
- I would like to see a better explanation of how U can be derived from crowd workers (top of page 5). I’m at a loss for what “we obtain  the binary reference U with each boundary” means. What are these boundaries? Are they different values for T? And how do you turn U at several boundaries into a single U?

minor points:
- I think in eq. (3) “i” is supposed to be “s”?

---

### Review · Reviewer_yiGy · 2025-11-06

**Summary Of Contributions:**

## Summary
- The paper’s core contributions are: (1) New Metrics for Safety Feedback Analysis – robust, non-parametric approaches (MPA and WRA) to quantify rater consistency and sensitivity to content severity; and (2) Empirical Insights in Multi-Cultural AI Safety Alignment – an application of these metrics on diverse datasets, revealing how demographic factors influence rating behaviors and providing guidance for curating more inclusive alignment datasets.

- This paper presents a novel data-driven framework for interpreting ordinal safety ratings from diverse human raters. The key contribution is the development of non-parametric *responsiveness to severity* metrics that quantify how consistently a rater (or group of raters) uses an ordinal scale (e.g. a 0–4 scale) in accordance with the underlying severity of AI safety violations. The authors define two complementary metrics, Monotonic Precision Area (MPA) and Weighted Recall Area (WRA), which together capture whether higher ratings correspond to objectively higher severity (monotonic ordering) and how well raters discriminate fine-grained severity differences. These metrics are combined (via a harmonic mean) into an overall responsiveness score, enabling comparison of different raters or demographic groups.

- Using these metrics, the paper analyzes two publicly available pluralistic safety feedback datasets as case studies. Dataset 1 (Rastogi et al., 2024) contains AI-generated image prompts with safety ratings from both expert raters (binary safe/unsafe labels) and crowd raters (Likert 0–4) spanning multiple demographic groups. Dataset 2 (Kumar et al., 2021) consists of ~107k toxic comments with 0–4 toxicity ratings by crowd raters of diverse religious, age, and sexuality backgrounds. By applying the proposed metrics, the authors uncover nuanced differences in how various demographic subgroups use the ordinal scales to express perceived severity of content. For example, in Dataset 1, certain ethnic groups (Latinx and East-Asian) show significantly lower MPA (less monotonic alignment with severity) compared to others, indicating their higher scores do not consistently correspond to higher severity as defined by expert guidelines. In Dataset 2, raters with strong religious affiliation or identifying as LGBTQ+ exhibit lower responsiveness (especially lower MPA) relative to the crowd consensus, while the youngest age group (18–24) shows the highest responsiveness (best capturing both broad and granular distinctions in toxicity). These findings demonstrate the metrics’ utility in extracting pluralistic insights about safety perception differences across cultures and identities.

## Strength
- Proposed Metrics Addressing a Critical Gap: The work introduces a novel set of metrics (MPA and WRA) specifically designed to handle pluralistic ordinal ratings. Unlike traditional correlation or ROC/AUC measures, these metrics account for how each score on the scale is used, enforcing monotonic relationship with severity and proper use of the full scale. This is a significant methodological advance for evaluating human feedback on safety – it directly tackles issues like rater bias, scale misuse, and non-monotonic judgment patterns that prior methods would miss. The paper also provides theoretical justification (statistical consistency proofs) and contrasts with alternatives (Spearman’s ρ, Kendall’s τ, IRT models, etc.), underscoring the robustness and novelty of the proposed approach.

- Empirical Evaluation and Insights: The authors demonstrate empirical rigor by applying their metrics to two distinct datasets (image safety and text toxicity) with different modalities, scales, and demographic axes. The analysis is comprehensive: they evaluate dozens of demographic subgroups (both intersectional groups like “Black–GenZ–Man” and top-level categories like ethnicity alone) and report statistical significance (using permutation tests and bootstrapped confidence intervals) for differences in responsiveness. The metrics successfully surface meaningful patterns – for instance, identifying that certain cultural groups (Latinx, East-Asian in Dataset 1) use the Likert scale in a less monotonic way relative to expert-defined severity, or that highly religious raters in Dataset 2 tend to give higher toxicity scores inconsistently with the general crowd’s view. Many of these findings align with or quantitatively deepen observations by the original dataset authors, lending credibility to the results. Overall, the paper provides valuable new insights into how demographic background affects safety perception – a topic of great importance for AI alignment in multi-cultural contexts.

- High Relevance and Potential Impact for AI Safety Alignment: The study addresses a timely problem in ML – how to ensure safety alignment data reflects diverse human values. By illuminating differences in safety feedback, the work has clear implications for relevance to TMLR’s audience (which includes researchers in ML fairness, human-AI interaction, and RLHF). The authors go beyond analysis and offer concrete guidance on using the metrics for improved data curation and model training. For example, they propose selecting the most responsive raters from each group and carefully including content where groups disagree, to build more inclusive reward models. They also caution against misusing the metrics to exclude groups, instead advocating to identify variations and adapt optimization objectives accordingly. This proactive discussion shows the practical significance of the work: it’s not just theoretical, but can directly inform better practices in collecting human feedback and training safer generative AI. Additionally, the authors emphasize reproducibility – they provide a Python implementation of the metrics in the appendix and use only open datasets, ensuring that others can apply and build on this work easily.

## Weakness
- Lack of Validation in Downstream Alignment Tasks: While the paper’s analysis is compelling, it stops short of demonstrating how using the proposed metrics actually improves AI models or safety outcomes. The authors recommend an action plan (e.g. filtering or weighting rater contributions based on responsiveness), but they do not implement this in a real alignment pipeline (such as training a reward model and showing reduced unsafe behaviors or false refusals). As a result, the work is primarily diagnostic – it reveals problems and suggests solutions, but does not prove that applying those solutions yields more robust aligned models. This limits the immediate practical evidence for the impact of the approach. A validation experiment (even a small-scale one) integrating these metrics into model training would strengthen the claims that more “responsive” feedback leads to better AI behavior.

- Dependence on a Chosen Reference Definition of Severity: The responsiveness metrics require a binary reference $U$ (which represents “ground truth” distinctions between lower vs higher severity) to evaluate against. In the case studies, $U$ is derived either from expert raters’ binary labels (Dataset 1) or from the aggregate crowd opinion (Dataset 1 & 2, via excluding the evaluated group). This raises a concern: if the reference itself is biased or incomplete, the metrics might flag a group as “unresponsive” when in fact the group has a legitimate perspective differing from the reference. For example, the paper finds that Black raters’ feedback is least aligned with the crowd’s consensus in one setting, which the authors note could be due to the crowd’s majority not fully reflecting the perspective of a historically marginalized group. This illustrates that responsiveness as defined by majority or expert consensus is not an absolute truth. The paper does acknowledge this nuance in its broader impact discussion, emphasizing that quantitative metrics alone cannot explain why such differences exist and cautioning against using them to simply exclude divergent groups. However, this inherent limitation remains: the insights are always relative to a particular chosen reference. The method may need careful adaptation if no reliable reference is available or if multiple, possibly conflicting, “ground truths” exist.

- Complexity and Clarity of Presentation: The technical derivation of MPA and WRA, while thorough, is quite dense and may be challenging for readers unfamiliar with ordinal statistics. The paper introduces several notations ($V, V_j, V_c, U,$ etc.) and two new metrics with formulae involving sums of precision/recall at each score. While the intuition is explained (e.g. “no monotonicity violations” vs “concordance at each score”), readers might still find it non-trivial to grasp and implement the metrics correctly. The reliance on appendices for simulations, implementation, and examples means the main text sometimes references important evidence (like simulation results or qualitative examples) without showing them. For instance, the paper claims traditional metrics can be misleading and refers to Appendix D for simulations, and mentions example items illustrating different group perceptions in Appendix F – these support points would have been stronger if summarized in the main paper. In a few cases, the text could be clearer: e.g., when first describing the binary reference U and how it’s obtained from crowd ratings, or when distinguishing “trisectional” vs “top-level” groups. Overall the writing is scholarly and detailed, but tightening the exposition (perhaps by adding a small diagram or example in the main text to illustrate how MPA/WRA are computed) would improve accessibility. This is a minor clarity issue; it doesn’t undermine the soundness of the work, but addressing it would help broaden the impact to practitioners.

**Audience:**

Yes

**Audience Explanation:**

Given the contributions, I feel people working on measurements of safety and evaluation of alignment would be interested in this work.

**Broader Impact Concerns:**

No concerns.

**Claims And Evidence:**

Yes

**Claims Explanation:**

- Overall, the paper’s claims are well-supported by evidence in the submission. The introduction and methodology claim that the new metrics capture both “broader distinctions and granular variations” in severity feedback – this is convincingly demonstrated through analysis of real rater behavior. For example, the results show cases where groups have comparable discrimination (WRA) but differ in monotonic ordering (MPA), indicating differences in granular use of intermediate scores. This validates that the metrics can disentangle broad vs fine-grained responsiveness as claimed. The authors also assert that traditional metrics (like correlation or AUROC) fail to capture certain biases or scale-use differences; evidence for this is provided via reasoning (e.g. two raters can have similar Kendall’s τ despite one using only a subset of the scale) and is further backed by simulations (referenced in Appendix D). While the full simulation results are in the appendix, the textual explanation in the main paper and the observed real-data trends (e.g. Kendall’s τ correlating with WRA but not reflecting monotonicity issues) strongly support the claim that conventional metrics are inadequate and that the proposed metrics offer a more nuanced, accurate lens.

- The claims about demographic differences in rating behavior – e.g. “safety perceptions are not uniform across groups” – are substantiated by statistically significant findings in both datasets. The paper frequently notes p < 0.05 for group differences and provides confidence intervals within ±0.01, indicating the results are not due to noise. Moreover, the authors contextualize these differences with prior work (for instance, confirming Kumar et al.’s observations about religious and LGBTQ+ raters with quantitative metrics). This cross-verification lends credibility to their interpretations. One claim in the abstract is that the approach can “improve the quality of pluralistic data collection and in turn contribute to more robust AI alignment”. While the analysis undoubtedly improves understanding of pluralistic data quality, the direct contribution to AI alignment is more qualitative at this stage – it’s argued through insights and recommended best practices rather than demonstrated via a trained model. The evidence supports that the metrics can inform better rater selection and feedback interpretation, but showing an actual alignment improvement is left as future work. This is a minor caveat. In sum, the paper backs up its technical and empirical claims with convincing data and analysis, and it is transparent about the claims’ scope (e.g. noting that their metrics alone do not explain why differences occur). The evidence provided is accurate and clear, positioning the reader to trust the results and conclusions drawn.

**Requested Changes:**

- Provide a Practical Demonstration of Impact: To strengthen the paper’s appeal, it would be crucial to include at least a small experiment or case study showing how using the responsiveness metrics can improve an AI alignment outcome. For example, the authors could train or simulate a reward model using a dataset curated with their suggested approach (selecting high-responsiveness raters from each group, etc.) and compare its performance to a baseline model trained on unfiltered data. Even a qualitative example of model behavior differences would substantiate the claim that these metrics lead to “more robust AI alignment.” This addition would directly support the paper’s high-level motivation by moving beyond analysis to impact. Without this, the work risks being seen as purely observational; demonstrating a tangible benefit would be critical for convincing readers of its significance in practice.

- Clarify the Definition and Use of the Reference Severity: Since the notion of “true” severity $V$ and the derived binary reference $U$ is central to the metrics, the paper should explain this setup more clearly in the main text. Readers might be confused about how $U$ is obtained (“guideline-based” vs “crowd-based”) and why multiple threshold boundaries are averaged for robustness. Clearly defining $V_g$ (expert-guided severity) vs $V_c$ (crowd consensus severity) and how $U$ is constructed from each would prevent misunderstanding. Additionally, discussing the limitations of these references (e.g., potential biases or when one should use expert vs crowd reference) in the main paper (not just Broader Impact) is important. This clarification is critical for correctness, because the interpretation of responsiveness scores hinges on understanding to what the raters are being responsive. A brief subsection in the Setup or Method explicitly on “Defining the Severity Reference” would resolve this issue.

---

### Review · Reviewer_sczV · 2025-11-30

**Summary Of Contributions:**

The paper investigates the challenge of interpreting ordinal safety ratings collected from diverse human raters. The authors propose a set of non-parametric responsiveness metrics Monotonic Precision Area (MPA) and Weighted Recall Area (WRA) to quantify the degree to which ordinal ratings reflect differing levels of underlying severity. Compared with non-parametric (e.g., AUROC, Kendall’s τ) or parametric IRT-style models, the proposed approaches make no assumptions about a common latent scale shared across raters, which is crucial in pluralistic settings. Experiments on two datasets demonstrate that the metrics reveal group-level variation that is not captured by previous methods.

**Audience:**

Yes

**Audience Explanation:**

The investigation in ordinal safety evaluation with heterogeneous cultural norms and rating behaviors is important, which could help the alignment process of LLM.

**Claims And Evidence:**

Yes

**Claims Explanation:**

- The author provide theoriatical proof for the proposed metrics, which makes the work more solid and convincing.
- The experimental analysis provides actionable insights into pluralistic data collection.

**Requested Changes:**

- While the author suggest that these metrics could help choose better raters or improve reward models, but it does not show any experiments proving that they actually improve alignment results. Adding small alignment experiments would make the contribution stronger.
- The structure of Section 5.2 could be improved. The key findings and insights should be presented more clearly to help readers understand the main takeaways.

---

### Decision · Action_Editor_QGzW · 2026-01-12

**Recommendation:** Accept as is

**Additional Comments:**

The primary critique of the paper leveled by all reviewers is that these metrics would be more effective if put into practice. I agree with this: the paper feels like a nice conceptual nugget, and one that future work can build on, but it will likely be effective if the authors themselves are able to do that building and show what this can do.

Beyond that, there are no required changes to the paper. It is clearly written and substantiates the claims it makes.

**Audience:**

Yes

**Audience Explanation:**

This paper is very timely and pluralistic alignment is an important topic. A number of readers would find it interesting.

**Claims And Evidence:**

Yes

**Claims Explanation:**

This paper claims to develop metrics that measure how different raters or rater groups respond to the severity of violations, and how responsive they are to changes in those levels of severity.  The two metrics are proven to obey desirable properties laid out in the paper. For analysis, the paper applies the metrics to two publicly-available pluralistic alignment datasets.  The claims are well supported by the evidence in the paper.